# Evolution of tropical cyclone genesis regions during the Cenozoic era

Qing Yan [1,2,3], Robert Korty[4], Zhongshi Zhang[5,6] & Huijun Wang[3]

How the substantial climate shifts of the Cenozoic era shaped the geographical distribution of tropical cyclone genesis remains unknown. Through a set of coupled model simulations, we demonstrate that conditions during the warmer Early Eocene are more favorable for storm formation over the Southern Hemisphere, particularly the South Indian Ocean. As the climate cools, there is an increasing favorability for genesis in the Northern Hemisphere and a coincident decrease in the Southern Hemisphere over time, with the locations most conducive to storms migrating equatorward in both hemispheres. A shift in the most favorable conditions to the western North Pacific likely occurs during the Pliocene, largely due to the closure of the tropical seaways, and marks the final establishment of modern tropical cyclone distribution. The substantial variations of genesis regions in the Cenozoic may affect upper-ocean vertical mixing and hence tropical/global climate, but are missed in most current deep-time simulations.

[1] Nansen-Zhu International Research Centre, Institute of Atmospheric Physics, Chinese Academy of Sciences, 100029 Beijing, China. [2] CAS Center for Excellence in Tibetan Plateau Earth Sciences, Chinese Academy of Sciences (CAS), 100101 Beijing, China. [3] Key Laboratory of Meteorological Disaster/ Collaborative Innovation Center on Forecast and Evaluation of Meteorological Disasters, Nanjing University of Information Science and Technology, 210044 Nanjing, China. [4] Department of Atmospheric Sciences, Texas A&M University, College Station, TX 77843, USA. [5] Department of Atmospheric Science, School of Environmental Studies, China University of Geosciences, 430074 Wuhan, China. [6] Uni Research Climate, Bjerknes Center for Climate Research, Bergen N-0007, Norway. Correspondence and requests for materials should be addressed to Q.Y. (email: yanqing@mail.iap.ac.cn) or to Z.Z. (email: zhongshi.zhang@cug.edu.cn)

Tropical cyclones (TCs) are one of nature's most destructive hazards and can result in catastrophic losses of life and massive economic losses over populated coastal regions. They respond to changes in climate and may also feedback on climate via modulating oceanic heat transport[1–3]. Today, TCs originate over tropical and subtropical oceans, biased towards the western sides of individual basins, and with more storms forming over the Northern Hemisphere than the Southern Hemisphere[4]. The western North Pacific accounts for approximately one third of all global TCs today and is the most productive storm basin in the world. This basic spatial distribution of TC genesis largely holds for simulations of climates with land-sea configurations similar to the present-day[5,6], even though there are important shifts poleward with warming[7,8].

During the Cenozoic era—spanning from ~65 Ma ago to present—the Earth's climate has undergone substantial changes, including a long-term cooling, which moved the planet from a hothouse world with ice-free poles, to an icehouse world with ice-covered poles[9]. Several important climatic events and transitions, some driven by tectonic plate motions and others by changes in greenhouse gas concentrations[10], created a rich diversity of climates during this era. Although there has been progress in our understanding of how storms may vary with climate during the Late Quaternary[11,12], the history of global TC activity during most of the Cenozoic remains unknown, owing to the dearth of paleo-storm records. Given the potential climate impact of TCs[3,13] and its dependence on the location of TC-induced mixing[13,14], it is of great interest to explore how the regions that support and sustain TCs may have varied during the large climate shifts of the Cenozoic—especially when and how the modern pattern of TC genesis was established.

Here we present a possible scenario for the Cenozoic evolution of TC genesis on tectonic timescales based on a suite of coupled climate model simulations. We find that conditions during the warmer Early Eocene are more conducive to storm formation over the Southern Hemisphere, with the South Indian Ocean being the most favorable basin. There is then an increasing favorability for genesis in the Northern Hemisphere and a coincident decrease in the Southern Hemisphere as the climate cools, alongside an equatorward shift in the most favorable conditions. In particular, we identify the important role of the closure of the tropical seaways during the Pliocene in establishing the modern-day TC distribution over the western North Pacific and Southern Indian Oceans.

## Results

**Genesis potential from the Early Eocene to present.** To investigate the Cenozoic TC evolution, we first simulate a wide range of Cenozoic climates using the low-resolution version of the Norwegian Earth System Model (NorESM-L), which includes the Early Eocene, Late Eocene, Late Oligocene, Early Miocene, Late Miocene, Late Pliocene, and today. We then adopt a genesis potential index (GPI), which summarizes changes in the large-scale environments that spawn and support storms, to estimate TC locations and frequencies, as low-resolution climate models are better able to reproduce these large-scale genesis factors than the TC-like systems themselves[15]. This method has proven useful in capturing the spatiotemporal variability of observed TC genesis[16,17](Supplementary Fig. 1) and has been widely applied to modeling studies of present-day, future, and paleoclimate periods[18–25].

Our study highlights that TC genesis varied substantially with the climate shifts of the Cenozoic. During the Early Eocene, simulated annual sea surface temperatures (SSTs) averaged over the tropics (30°S–30°N) reach ~30 °C (5 °C higher than in the pre-industrial), resulting from high $CO_2$, and within the range suggested by proxy-based estimates[26]. Although the 26 °C isotherm is often used in modern meteorology to highlight areas with favorable thermodynamic conditions for TC production and high potential intensity, this indicator is not useful in other climates. The isotherm is displaced poleward to ~40°N/S during the Early Eocene (Fig. 1a), but many locations enclosed by it are shown to be much less favorable for storm formation than today. This is attributed to the fact that the isotherm separating regions that support deep convection (e.g. potential intensity > 50 m/s[27,28]) from those that do not, varies with climate. For example, relative to the present-day, the North Pacific and North Atlantic in the Early Eocene were much less favorable for TC genesis in the deep tropics, whereas some regions with conditions that support storms appear at higher subtropical latitudes in the central North Pacific (Fig. 1a, g). During the Early Eocene, when the Australian continent was confined south of ~30°S, there was a zonal band with favorable conditions for genesis extending from ~30°E to 120°W across the Southern Hemisphere tropics. The South Indian Ocean was the basin most conducive to storm formation during this period, rather than the western North Pacific.

We consider this reorganization of potential TC distribution in the Early Eocene to be due to the adjustment of the atmospheric thermal structure and its associated circulations. Over the western North Pacific, the decreased potential intensity in the Early Eocene at low latitudes (5–12°N) is mainly caused by the reduced air-sea disequilibrium relative to present (Fig. 2b) resulting from weakened surface radiative flux and enhanced surface wind speed (Supplementary Fig. 2), whereas it is largely attributed to higher outflow temperatures at higher latitudes, which leads to smaller thermodynamic efficiency given larger warm temperature anomalies in the upper troposphere than the surface (Supplementary Fig. 3). The meridional temperature gradient in the troposphere is intensified south of ~22°N and leads to enhanced vertical wind shear in thermal wind balance (Fig. 2c); the reduced temperature gradient contributes to the weakened wind shear at the north of ~22°N. The moist entropy deficit anomaly broadly exhibits a tripole pattern, with a decrease at the zonal band of ~12–20°N and an increase along its northern and southern sides (Fig. 2d). The increased moist entropy deficit is largely caused by a decreased temperature contrast between the mid-troposphere and the surface, which weakens the strength of the surface heat fluxes, whereas the smaller deficit between 12–20°N results from increased relative humidity (Fig. 2d). The lower potential intensity, enhanced wind shear, and larger moist entropy deficit all point to decreasing favorability for TC genesis in the Early Eocene over the deep tropics (~ 5–15°N) of the western North Pacific (Fig. 2a).

In contrast, all of the genesis factors point to more favorable conditions for storm generation in the Early Eocene over the South Indian Ocean, except equatorward of 10°S (Fig. 2e-h). The higher potential intensity south of 10°S is attributed to the increased air-sea disequilibrium because of the weakened surface wind speed and larger thermodynamic efficiency owing to the lower outflow temperature (Fig. 2f and Supplementary Fig. 4). The reduced wind shear between ~12–26°N results from decreased meridional temperature gradient that leads to a smaller vector difference in the winds between the upper and lower troposphere (Fig. 2g). The decreased moist entropy deficit over the majority of South Indian Ocean results from the larger relative humidity, though the vertical temperature contrast is also decreased (Fig. 2h). The favorability of TC formation is also decreased over the entire eastern North Pacific, arising largely from enhanced wind shear and larger moist entropy deficits, while favorable conditions shift poleward in both the Southern Pacific and North Atlantic (Supplementary Fig. 5), consistent

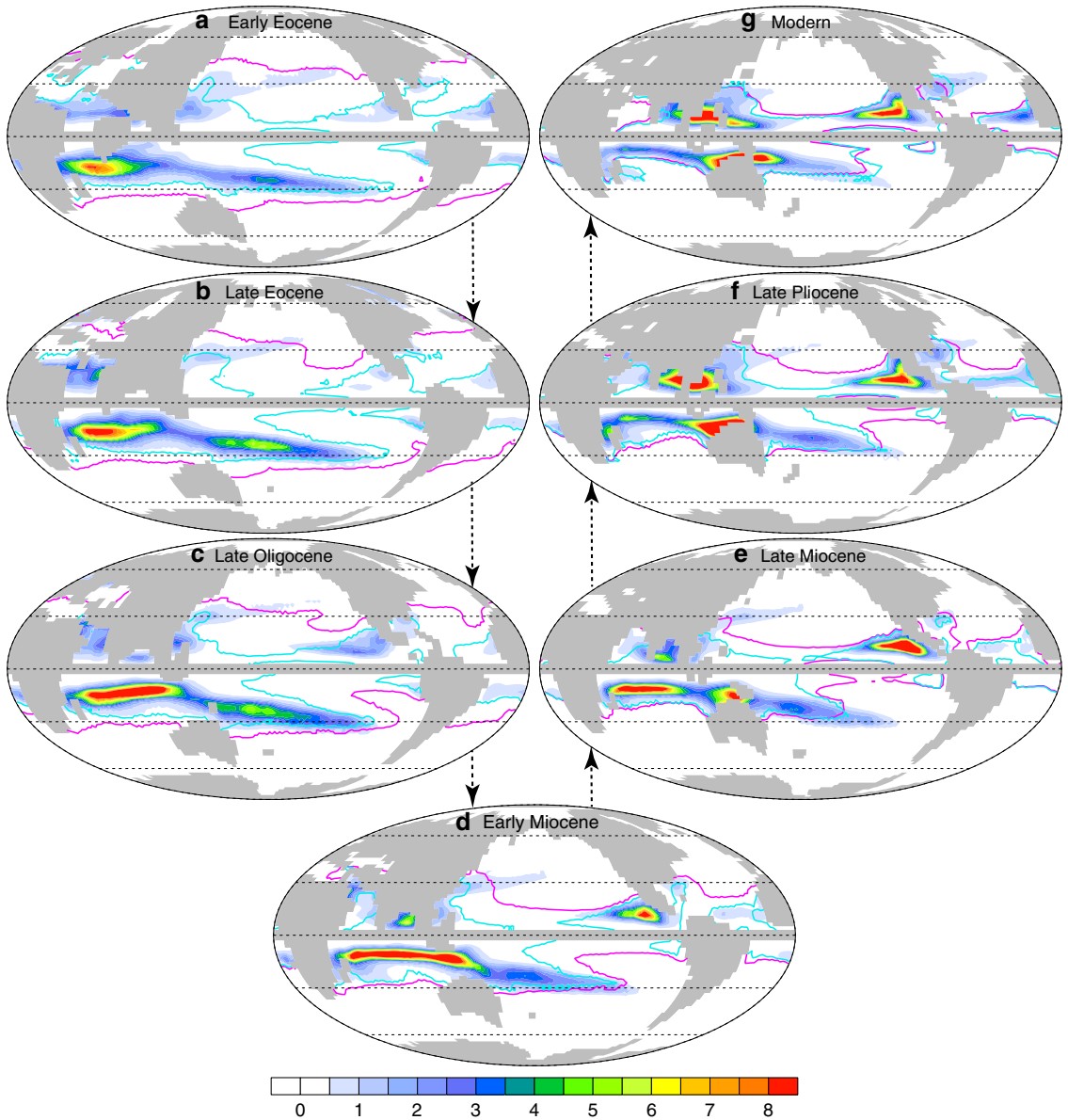

**Fig. 1** Storm season mean genesis potential distribution during the Cenozoic era. **a** Early Eocene, **b** Late Eocene, **c** Late Oligocene, **d** Early Miocene, **e** Late Miocene, **f** Late Pliocene, and **g** today. The pink lines show the 26 °C isotherm. The cyan lines show the 70 m s$^{-1}$ contour for potential intensity. The 26 °C isotherm is collocated with the boundary of regions with mean potential intensity of 70 m s$^{-1}$ in the modeled present climate, but the isotherm that correlates with high potential intensity varies across climates. Storm season is defined as Aug–Sep–Oct for the Northern Hemisphere and Jan–Feb–Mar for the Southern Hemisphere. The units of genesis potential are events m$^{-2}$ month$^{-1}$(×10$^{-13}$)

with a prior study[5]. Overall, favorable conditions for genesis in the Early Eocene shift to higher tropical latitudes and away from very low ones, which may be tied to the poleward migration of the Hadley circulation in the Early Eocene relative to present (Supplementary Fig. 6).

As the climate cools through the Cenozoic, conditions across the Northern Hemisphere become more favorable with time, while the Southern Hemisphere becomes less favorable later in the era (Fig. 3a, e). The latitude of peak GPI broadly shifts equatorward in both hemispheres over the era. However, the environments supporting storm genesis over time are very different between ocean basins. Over the western North Pacific (Fig. 3b), conditions remain relatively unfavorable for genesis from the Early Eocene through the Late Miocene, but this is followed by a remarkable increase and equatorward shift in the GPI at the transition to the Pliocene. Broadly similar behavior is

seen in the North Atlantic (Fig. 3d). The GPI over the eastern North Pacific also increases during the Cenozoic, but actually begins to rise in earlier time-intervals, peaking in the Late Miocene (Fig. 3c). In the Southern Hemisphere (Fig. 3e–h), the GPI is more varied by region, but there is a large decrease in the latitude of the peak GPI—shifting nearly 10° equatorward from the Early Eocene to the present-day over each ocean basin.

**The Late Miocene-Pliocene transition.** One of the most important transitions in the evolution of the GPI is the establishment of a favorable region for genesis centered over the western North Pacific, with a concurrent reduction in favorability over the South Indian Ocean (Figs. 1, 3). The timing of this transition must have occurred after the Late Miocene and before the Late Pliocene (LMLP; ~10–3 Ma), given the similarity in

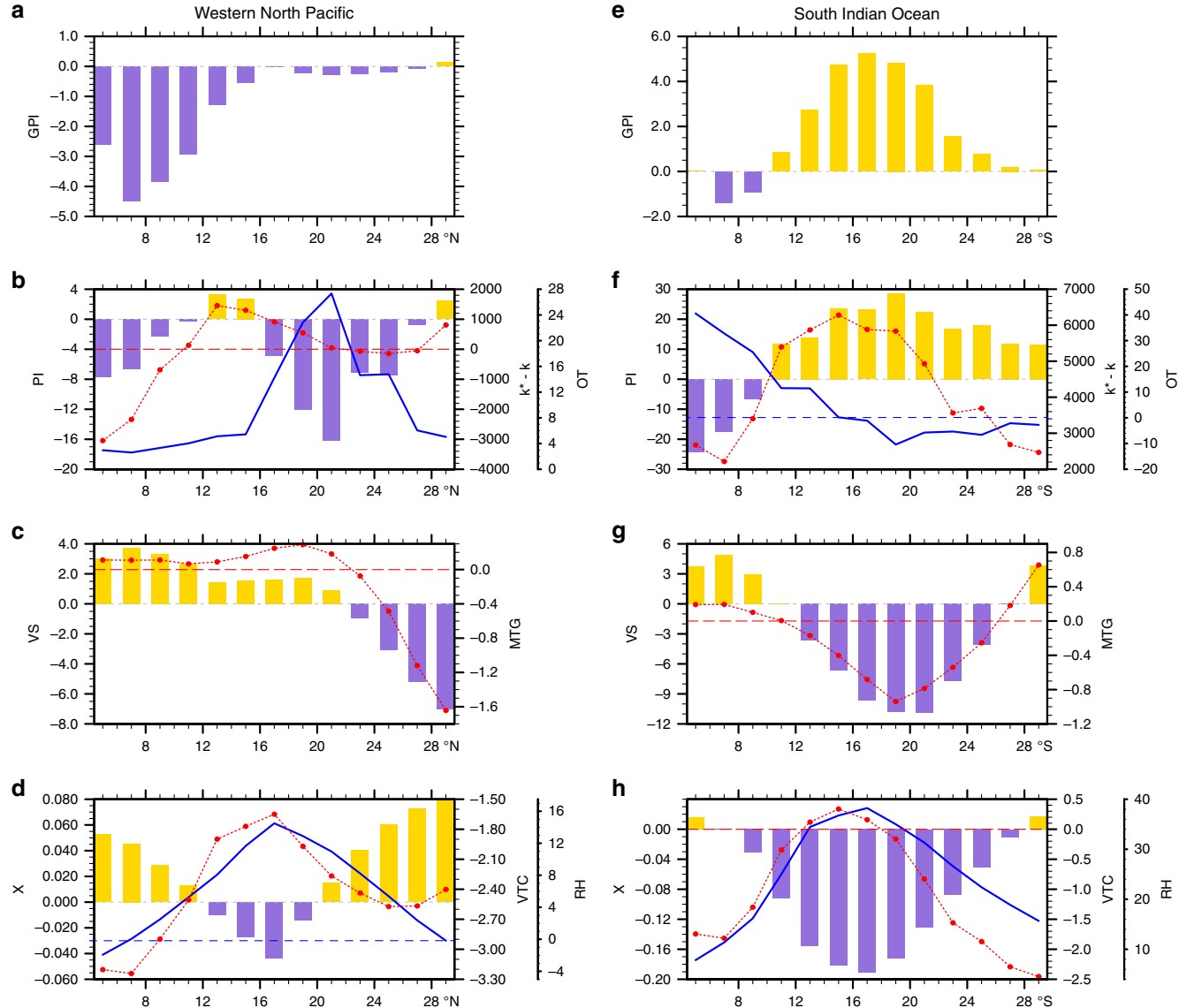

**Fig. 2** Differences in zonal mean genesis potential and environmental variables during storm season between the Early Eocene and pre-industrial. Over the western North Pacific (110–160°E): **a** Genesis potential index (GPI; number of events per month). **b** Potential intensity (PI; bars; m s$^{-1}$), the enthalpy difference between sea surface and boundary layer (k*–k; red line; J kg$^{-1}$), and the outflow temperature (OT; blue line; °C). **c** Vertical wind shear between 200 and 850 hPa (VS; bars; m s$^{-1}$) and absolute meridional temperature gradient (MTG; red line; ×10$^{-6}$ °C m$^{-1}$) in the troposphere. **d** Moist entropy deficit (X; bars), vertical temperature contrast (VTC; red line; °C) between surface and mid-troposphere (600 hPa), and relative humidity (RH; blue line; %) at 600hPa. **e–h** same as **a–d** but for the South Indian Ocean (40–100°E)

global TC distribution between the Late Pliocene and the present-day (Fig. 1).

What causes this shift in the TC genesis potential during the LMLP transition? Tectonic motions and changes in $CO_2$ concentration are two potential factors, as these are the two most obvious differences in boundary conditions between the Late Miocene and Late Pliocene experiments and can exert a significant influence on tropical climate. Sensitivity experiments indicate that the geographic distribution of genesis potential in the Late Miocene is broadly insensitive to the $CO_2$ level applied (i.e., 350 vs. 560 ppmv), and the pattern of modern genesis regions largely remains unchanged in response to rising $CO_2$ (Supplementary Fig. 7). Thus, the LMLP transition in TC genesis may be largely attributed to changes in the position of the continents—among which the restriction of Indonesian seaway and the closure of Panama seaway stand out.

To test the effect of the tropical seaway closures in potential TC formation, we perform two sensitivity experiments with the NorESM-L. Taking the Late Pliocene experiment with closed Indonesian and Panama seaways as the baseline (referred to as LP_closeIP), in the first experiment we keep the Indonesian seaway closed but open the Panama seaway (referred to as LP_closeI); in a second experiment (referred to as LP_noclosure), we open both tropical seaways. The results indicate that the closure of the tropical seaways leads to higher temperatures over the western North Pacific with weakened surface wind speed (i.e., LP_closeIP minus LP_noclosure; Fig. 4a) and cooling in the South Indian Ocean (except at very low latitudes) with enhanced surface wind speed—changes that are supported by geological evidence[29–31]. Higher ocean heat convergence in the equatorial Pacific leads to higher SSTs, and these contribute directly to larger potential intensity here, with additional support from decreases in

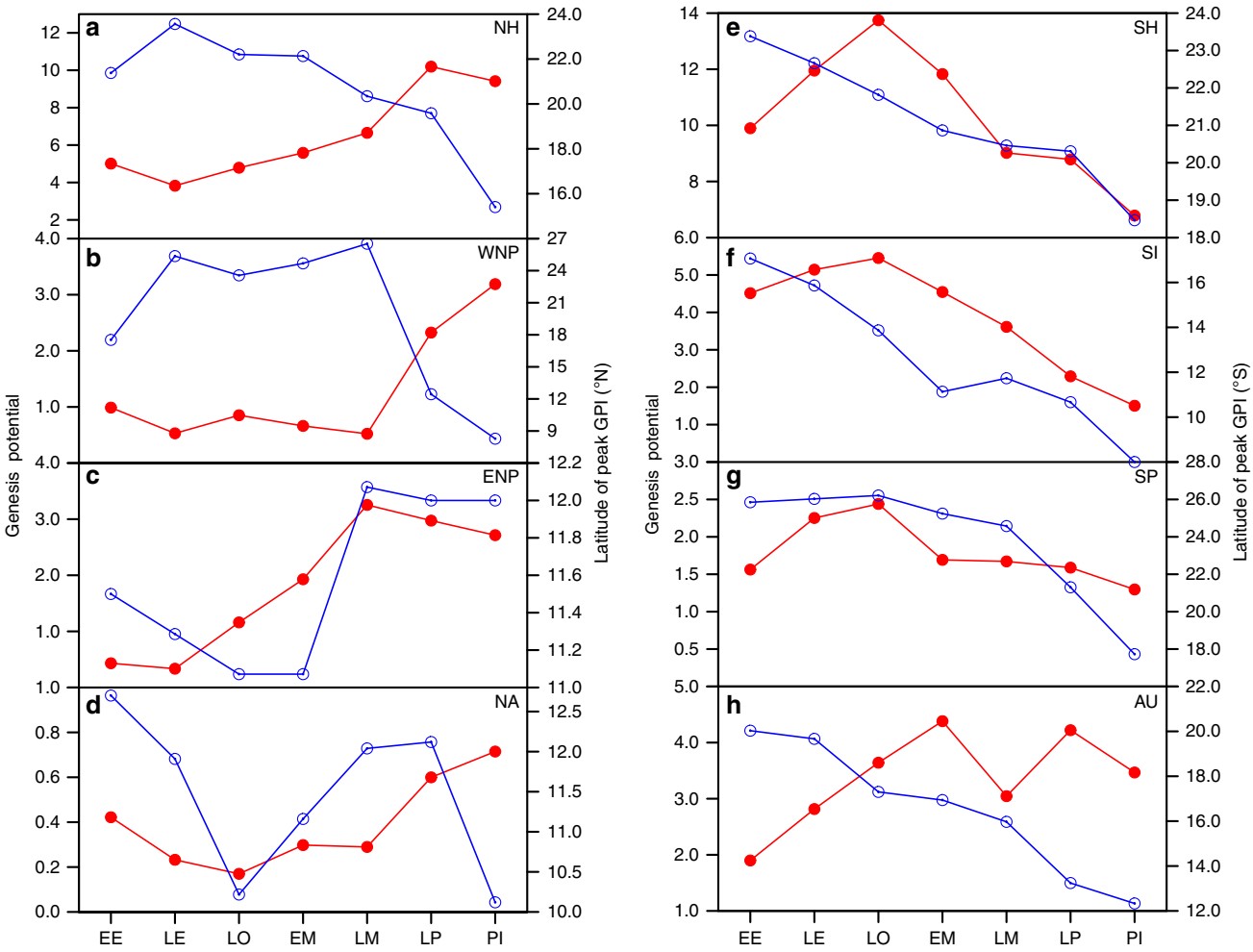

**Fig. 3** Evolution of storm season genesis potential (red solid circle) and the latitude of peak genesis potential (blue hollow circle) averaged over individual storm basins. **a** Northern Hemisphere (5–45°N), **b** Western North Pacific (5–30°N; 110–160°E), **c** Eastern North Pacific (5–30°N; 150–95°W), **d** North Atlantic (5–20°N; 60–10°W), **e** Southern Hemisphere (SH; 5–45°S), **f** South Indian Ocean (5–30°S; 40–100°E), **g** Southern Pacific (5–30°S; 165°E–130°W), and **h** Australian coast (5–30°S; 100–165°E). EE Early Eocene, LE Late Eocene, LO Late Oligocene, EM Early Miocene, LM Late Miocene, LP Late Pliocene, PI pre-industrial. For the latitude of peak genesis potential, we first find the latitude of the maximum genesis potential across each longitude over a storm basin and then compute the mean of the selected latitudes. The units of genesis potential are number of events per month

surface wind speeds in the western end of the basin near Indonesia (Fig. 4b and Supplementary Figs. 8, 9). Conversely, decreases in ocean heat convergence in the Indian Ocean between 10° and 20°S lead to surface cooling and reductions in potential intensity, with contribution from enhanced surface wind speeds. Vertical wind shear is reduced over the western North Pacific, consistent with the weaker meridional temperature gradient in the troposphere, whereas it exhibits an increase south of 10°S over the South Indian Ocean (Fig. 4c, d). Moist entropy deficits rise over the South Indian Ocean, which arises from decreased mid-tropospheric relative humidity, consistent with anomalous descent (Fig. 4e, f). In the equatorial Pacific, moist entropy deficits are lower, mainly as a result of larger relative humidity from stronger ascent. Each of these factors contributes to a more favorable environment over the western North Pacific, but a less favorable one in the South Indian Ocean (except at very low latitudes where GPI is increased) when the tropical seaways are closed (Fig. 4g), and there is an overall equatorward migration of genesis regions.

The spatial changes in the genesis regions induced by the tropical seaway closures (Fig. 4g) are qualitatively consistent with the differences between the Late Miocene and Late Pliocene over

the western North Pacific and South Indian Ocean (Supplementary Fig. 10). These results suggest the role of tropical seaway closures in the LMLP transition dominates in terms of genesis distribution. Further, the restriction of the Indonesian seaway is more important in the reduced genesis potential over the Southern Indian Ocean, while the closure of the Panama seaway has a larger effect on increasing favorable conditions over the western North Pacific. The narrowing Indonesian seaway affects the large-scale environmental conditions important to storm formation over the western Pacific (Supplementary Fig. 11) via the regulation of the Indonesia Throughflow (Supplementary Fig. 12). However, these effects are more profound over the equatorial Pacific in our simulations, where storms rarely form as the Coriolis parameter is weak. In contrast, the closure of the Panama seaway strongly affects the genesis potential over the western North Pacific, and shows limited influence over the low latitudes of the South Indian Ocean (Supplementary Fig. 13).

## Discussions
The closure of tropical seaways accounts for most of the spatial changes in GPI between the Late Miocene and Late Pliocene, but

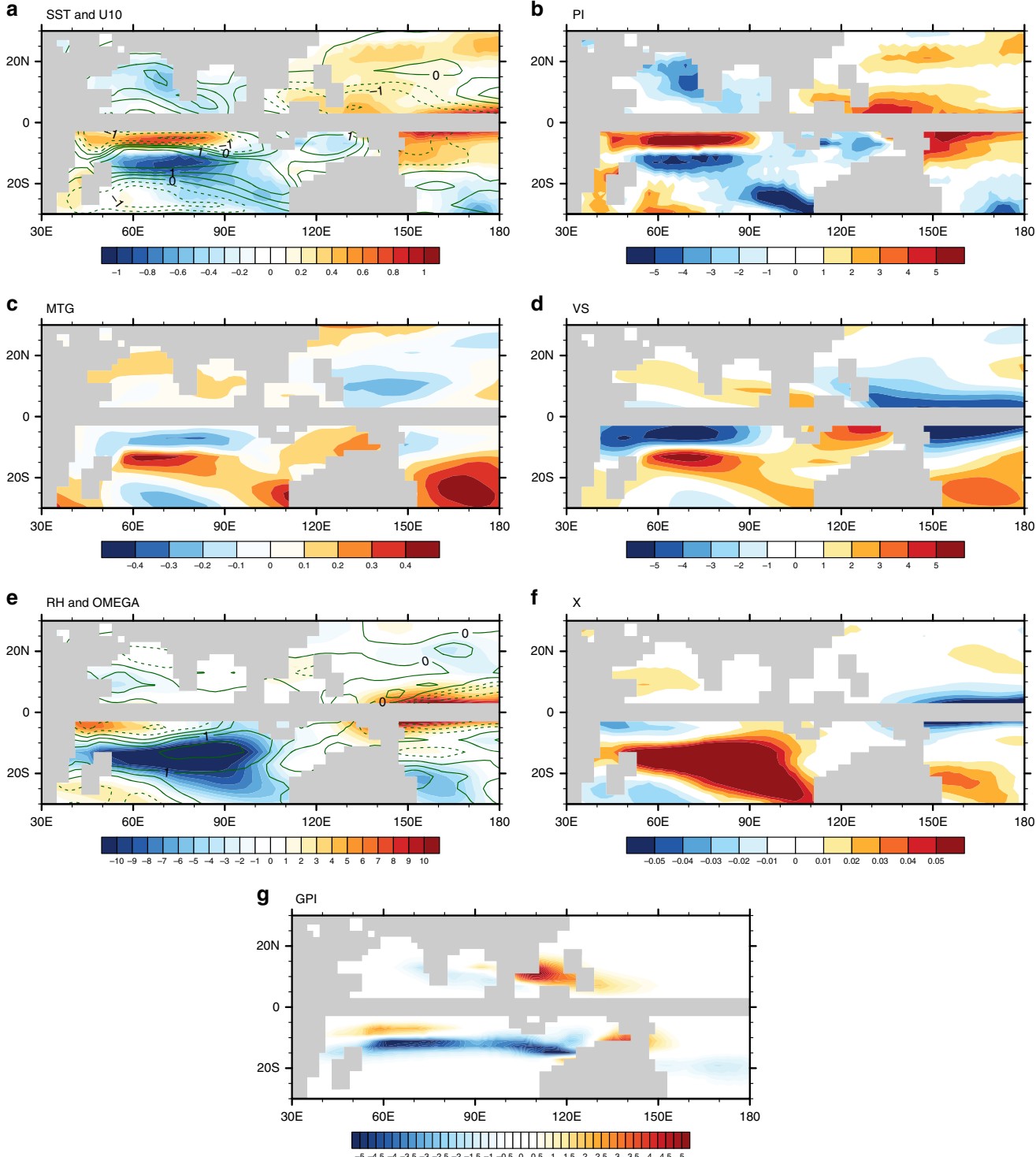

**Fig. 4** Differences in large-scale environmental conditions during storm season caused by the tropical seaway closures (LP_closeIP minus LP_noclosure). **a** Sea surface temperature (shading; °C) and surface wind speed (contour; m s$^{-1}$). **b** Potential intensity (m s$^{-1}$). **c** Absolute meridional temperature gradient ($\times 10^{-6}$ °C m$^{-1}$) in the troposphere. **d** Vertical wind shear between 200 and 850 hPa (m s$^{-1}$). **e** Relative humidity (shading; %) and vertical velocity (contour; Pa s$^{-1}$) at 600 hPa. Negative and positive values in vertical velocity show anomalous ascents and descents, respectively. **f** Moist entropy deficit. **g** Genesis potential ($\times 10^{-13}$ events m$^{-2}$ month$^{-1}$)

the amplitude is underestimated compared to the modeled GPI changes during the LMLP transition (Supplementary Figure 10). This may be attributed to the relatively smaller depth and width of tropical seaways applied here, which largely affect the magnitude of the modeled climate response[32,33]. Additionally, the

timing for TC development is not synchronous over each ocean basin, and the associated mechanisms may be different in each (Figs. 1, 3). The Cenozoic evolution of genesis potential over the South Indian Ocean is a more progressive process than the abrupt transition that occurs in the western North Pacific, which may be

linked with the gradual narrowing of Hadley cell in the Southern Hemisphere (Supplementary Fig. 14) due to the northward incursion of Australia.

Though TCs have been examined in only a few other deep-time simulations, our results show some qualitative similarities with other models. For example, GPI is lower in the western North Pacific during the Late Pliocene relative to present-day in our model, which is similar to the ensemble mean considered in the Pliocene Model Intercomparison Project[20]. The decreased favorability for genesis at low latitudes in the Early Eocene is also consistent with TCs downscaled from an Eocene-like climate simulation[5]. These similarities provide some confidence that our results may be model-independent, but prior work has shown differences among models at basin scales[18,20]. The precise composition of genesis indices can also affect results, but we show that the main qualitative features and temporal pattern reported here are insensitive to the specific GPI formula employed (Supplementary Fig. 15).

Notably, proxy records are scarce or absent for constructing the spatial distribution of TC activity during most of the Cenozoic, and the GPI provides little information on TC tracks and landfalls. Therefore, the model results cannot be robustly evaluated by geological evidence. Nevertheless, our study provides a testable relationship between tectonic/$CO_2$-induced climate change and TC behavior, and sheds light on TC responses to very high $CO_2$ levels that may be seen in the future if fossil-fuel emissions continue unabated[34]. Moreover, the substantial variations in genesis regions in most of Cenozoic imply potential changes in the TC-induced vertical mixing in terms of both magnitude and spatial distribution (Supplementary Figs. 16, 17). Given the role that TCs could play in modulating tropical oceans and possibly global climate[3,13,35], we highlight that future modeling studies targeting deep-time simulations should consider the feedback of TCs associated with upper-ocean vertical mixing.

## Methods

### Introduction to NorESM-L.
The low-resolution version of the Norwegian Earth System Model (NorESM-L) is a fully coupled climate model developed for paleoclimate simulations[36,37] at the Bjerknes Centre for Climate Research. NorESM-L is built under the structure of the Community Earth System Model and consists of four components coupled together. The atmospheric component is the Community Atmospheric Model version 4 (CAM4), which has a horizontal resolution of ~3.75° × 3.75° with 26 vertical levels. The land component is the Coummunity Land Model version 4 and adopts the same horizontal resolution as CAM4. The ocean and sea-ice component is the Miami Isopycnic Coordinate Ocean Model (MICOM) and Los Alamos Sea Ice Model version 4 (CICE4), respectively. MICOM has a nominal 3° horizontal resolution and 32 vertical levels. The model has been proven to be skillful in capturing the present-day climate and performs well in reproducing the majority of features across a wide range of paleoclimates[38–40].

### Experiment design for simulating Cenozoic climates.
In this study, we first perform seven coupled NorESM-L experiments to simulate the climate of the Early Eocene, Late Eocene, Late Oligocene, Early Miocene, Late Miocene, Late Pliocene, and today (Supplementary Table 1). Each experiment is integrated for 2200 model years, and we analyze the outputs of the last 200 years.

The continental configurations for the Early Eocene, Late Eocene, Late Oligocene, Early Miocene, and Late Miocene are based on the reconstructed paleogeographic maps[41] for 50, 40, 30, 20, and 10 Ma, respectively (Supplementary Fig. 18). These paleogeographic maps include information on ancient mountain ranges and shorelines, active plate boundaries, and the extent of paleoclimatic belts. We set the mountain lines and coastlines at 1000 m and 0 m, respectively, although uncertainties in estimating paleo-altitude still exist. The shallow ocean basins are set to a depth of 200 m. All these altitude and depth data are interpolated to a 1° × 1° resolution, and then are used to construct the paleo-topography and paleo-bathymetry used in NorESM-L. Regarding the Late Pliocene, we use the reconstructed topography from the Pliocene Research, Interpretation and Synoptic Mapping Project (PRISM3D)[42] with modern land-sea mask, following the guidelines of the Pliocene Model Intercomparison Project[43].

For the greenhouse gases, we set atmospheric $CO_2$ concentration to 1120 ppmv for the Early Eocene (Supplementary Table 1), which corresponds to the upper estimate according to a previous synthesis[44]. $CO_2$ concentration is set to 1050

ppmv for the Late Eocene, 700 ppmv for the Late Oligocene, 420 ppmv for the Early Miocene, 350 ppmv for the Late Miocene, and 405 ppmv for the Late Pliocene. The other greenhouse gases (e.g., $CH_4$ and $N_2O$) are fixed at the pre-industrial levels.

Given the scarcity of proxies for vegetation during the Cenozoic era, we choose to adopt the same idealized land cover for each of these paleoclimate simulations, except for the Late Pliocene for which we adopt the PRISM3D reconstructions. Specifically, forest is prescribed between 30°S and 30°N, with shrub and grass outside this latitude band. There are no ice sheets on Antarctica and Greenland.

### Sensitivity experiments for tropical seaway closure.
Geological evidence suggests that the tropical seaways of Indonesia and Panama closed during the Pliocene (~5–3 Ma)[29,45], preventing surface water exchange between the Pacific and Indian Oceans and the Atlantic—although the exact timing of these closures remains unclear[46]. Taking the aforementioned Late Pliocene experiment as the reference run (referred to as LP_closeIP), we perform two sensitivity experiments to examine the potential role of tropical seaway closures (Supplementary Table 2). In one sensitivity experiment, we open both the Indonesian and Panama seaways (referred to as LP_noclosure), leaving the other boundary conditions unchanged from the Late Pliocene run. Specifically, the Indonesian seaway is broadened by converting the northern part of New Guinea (11 grid cells) to ocean with a depth of ~50 m (Supplementary Fig. 19). The Panama seaway is opened by removing one land grid cell and setting the depth of new ocean grid cell to 25 m (Supplementary Fig. 19); we verify there are through-flows between the Atlantic and Pacific under this scenario. In the other sensitivity experiment, only the Panama seaway is opened (referred to as LP_closeI). The modifications of tropical seaways in these sensitivity experiments are designed to roughly represent the Pliocene palaeoceanographic conditions. It should be noted that the modeled climate responses may be dependent on the depth and width of tropical seaways applied in the model[32,33]. Each experiment is integrated for 1500 model years, and we analyze the output of the last 200 years.

### Genesis potential index.
In this study, we adopt the Genesis Potential Index (GPI) to summarize changes in the large-scale environments that spawn and support TCs. It is defined as:[18,47,48]

$$\text{GPI} = \frac{a[\min(|\eta|, 4 \times 10^{-5})]^3 [\max(\text{PI} - 35, 0)]^2}{\chi^{4/3}(25 + \text{VS})^4},$$ (1)

where $a$ is a normalizing coefficient, $\eta$ is the absolute vorticity at 850 hPa, PI, VS, and $\chi$ are the potential intensity, vertical wind shear, and moist entropy deficit, respectively (see below for details). Preliminary evaluation shows that the GPI simulated with NorESM-L reproduces the spatial pattern of present-day TC genesis well (Supplementary Fig. 1). These environmental factors have proven useful in predicting the areas where TCs form using a statistical downscaling method[22,48].

Potential intensity measures the thermodynamically achievable intensity of a storm and is defined as:[49]

$$\text{PI} = \sqrt{\frac{C_k}{C_d} \frac{\text{SST} - T_o}{T_o}(k_0^* - k)},$$ (2)

where $T_o$ is the mean outflow temperature, $C_k$ is the exchange coefficient for entropy, $C_d$ is the drag coefficient, $k_0^*$ is the enthalpy of air saturated at the sea surface temperature and pressure, and $k$ is the enthalpy of an ambient boundary layer parcel. The thermodynamic efficiency is measured by the term involving SST and $T_o$, and is mainly determined by the mean outflow temperature (approximately equivalent to the ambient tropopause temperature in the deep tropics)[50]. The difference between enthalpy of a saturated parcel and that of the ambient environment represents the thermodynamic disequilibrium at the sea surface[50]. Regions of high potential intensity identify the areas in which TC genesis is possible; low values are found only in regions that cannot support the deep convection that characterizes such systems[18]. Note that we modify the original algorithm by allowing it to scan soundings to 10 hPa, which allows for the possibility that convection may penetrate to much higher altitudes in warmer climates than it does today[5].

The reasons behind differences in potential intensity can be evaluated by considering the surface energy balance. In a steady-state in which the ocean's mixed layer remains approximately in thermal equilibrium, the thermodynamic disequilibrium term in Eq. 2 can be written as:[51]

$$k_0^* - k = \frac{F_{\text{net}} + F_{\text{ocean}}}{C_k \rho |\mathbf{V}|},$$ (3)

where $\rho$ is an average air density near the surface, $|\mathbf{V}|$ is the average surface wind speed, $F_{\text{net}}$ is the difference between the net solar flux into the ocean and the net infrared radiative flux out of the ocean, and $F_{\text{ocean}}$ is the net energy flux into the ocean mixed layer by ocean processes. Substituting Eq. 3 into Eq. 2 yields an alternate expression for PI that can be used to assess the underlying physical causes

behind changes in the quantity:

$$PI = \sqrt{\frac{SST - T_o}{T_o} \frac{F_{net} + F_{ocean}}{C_d \rho |\mathbf{V}|}}, \tag{4}$$

Squaring both sides of Eq. 4 and taking the natural logarithm allows contributions from individual terms in the PI formula to be isolated:

$$2\ln(PI) = \ln\left(\frac{SST - T_o}{T_o}\right) + \ln(F_{net}) + \ln(1 + \frac{F_{ocean}}{F_{net}}) - \ln(C_d \rho |\mathbf{V}|), \tag{5}$$

Supplementary Figs. 8 and 9 show the contributions of individual terms. Note that for the purposes of this partitioning, we consider annual mean quantities rather than storm season extremes for two reasons. First, Eq. 3 is valid on long timescales for which the ocean has reached equilibrium. Additionally, potential intensity itself exhibits only a weak annual cycle equatorward of ~20° latitude.

Vertical wind shear broadly hampers TC genesis and intensification by shearing the convective towers and ventilating the storm's core with sub-saturated air[52]. Here we define vertical wind shear as the magnitude of the vector difference between the 200 and 850 hPa horizontal wind vectors.

Moist entropy deficit is used to measure the mid-tropospheric moisture content[22]:

$$\chi = \frac{s^* - s_m}{s_0^* - s^*}, \tag{6}$$

$$s = c_p \log(T) - R_d \log(p_d) + \frac{L_{vo} r_v}{T} - R_v r_v \log(RH), \tag{7}$$

where $s_m$ is the mid-tropospheric moist entropy (at 600 hPa), and $s_0^*$ and $s^*$ are the saturation moist entropies of the sea surface and free troposphere (evaluated at 600 hPa), respectively. In Eq. 7, $T$ is the temperature, RH is the relative humidity, $p_d$ is the partial pressure of dry air, and the other parameters are constant. Larger values of the moist entropy parameter are indicative of less favorable conditions for storm formation, all other things being equal.

Low-level absolute vorticity (at 850 hPa) gives rise to important synoptic convergence around which convective systems can begin to organize[53]. This quantity is incorporated in some genesis indices, but others have argued it is not rate limiting far from the equator[54].

To test the sensitivity of our results to the GPI formula, we employ another genesis index[54]:

$$GPI_{TCS2011} = \exp(b + b_\eta \min(\eta, 3.7 \times 10^{-5}) + b_{shear} V_{shear} + b_{RH} RH + b_T T + \log(\cos\phi)), \tag{8}$$

where $\phi$ is the latitude and $T$ is relative SST; $b = -5.8$, $b_\eta = 1.03$, $b_{RH} = 0.05$, $b_T = 0.56$, and $b_{shear} = -0.15$. The qualitative results reported here, (i.e. equatorward shift of the locations of potential storms in both hemispheres during the Cenozoic and a shift of the most favorable conditions to the western North Pacific during the Pliocene), are largely independent of the GPI formula used (Supplementary Fig. 14).

## Data availability

The modelled genesis potential during the Cenozoic era (Fig. 1) is available at https://pan.cstcloud.cn/s/Z6kaMlxGSoU. All the climate model outputs are available from the corresponding author upon reasonable request.

## Code availability

The code for calculating potential intensity is provided by Prof. Kerry Emanuel and is available at ftp://texmex.mit.edu/pub/emanuel/TCMAX/. The code for NorESM model is available at https://wiki.met.no/noresm/start. The codes used for the numerical procedures are available from the corresponding author upon request.

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

## Acknowledgements

This study was funded by the National Natural Science Foundation of China (41888101, 41772179), the Youth Innovation Promotion Association by CAS (2019080), and the Young Elite Scientists Sponsorship Program by CAST (2017QNRC001). Z.Z. thanks the support of Thousand Talents Program for Distinguished Young Scholars. Language editing was performed by E.J. Farmer Editing.

## Author contributions

Q.Y. and Z.Z. designed the study. Q.Y. performed analyses. Z.Z. carried out the NorESM-L simulations. Q.Y. wrote the initial draft of the manuscript, with inputs from R.K. and H.W. All authors discussed the results and reviewed the manuscript.

## Additional information

**Competing interests:** The authors declare no competing interests.

