## [Peer Review File · Nature Communications]

Reviewers' comments:

Reviewer #1 (Remarks to the Author):

Review of "Evolution of tropical cyclone genesis regions during the Cenozoic Era" by Chris Brierley (UCL)

I am impressed by this manuscript by Yan et al and feel that it is worthy of publication. They have undertaken a novel piece of research by combining the (sub)disciplines of palaeoclimatology and tropical meteorology to demonstrate an interesting finding about the earth system. I naturally have some questions and comments. But whether these are sufficient to require revisions, I will leave up to the editor to decide.

Methodologically, it builds upon established approaches and therefore is sound. NorESM-L is a climate model specifically built for past climate research. It was part of the Pliocene Model Intercomparison Project, so its fidelity at simulating warm climates can be assessed. It generally shows a similar behaviour to the other climate models. The use of a genesis potential index is also a well-established approach. Results can vary depending on the precise composition of the GPI, but the authors demonstrate that this is not a problem in this situation. I was somewhat surprised that there were not a few words comparing the NorESM-L late Pliocene simulated GPI change with the other PlioMIP models (from my 2015 paper). This is only because an Indian Ocean intensification only occurs in the MIROC5 model, and not the other 4 models we looked at (NorESM-L was not investigated in that paper). The late Pliocene is only a weak signal compared to the other setups testing here, so one would expect a weaker less robust signal. I do not want to see the point of model-dependency belaboured to the detriment of this interesting research. But since there is some pertinent evidence for it, I feel it should be acknowledged in the text.

My other substantive comment is that I wondered why you concentrate on the Indonesian Seaway as a being something like a tipping-point with an open vs closed state. Looking at the gradual reduction in GPI throughout the Cenozoic (Fig 3f), I got the impression this a more progressive process due the gradual incursion of Australasia into the Tropics. I suspect that there is a more to be found by looking at the tropical circulations, but this could easily form the basis of a paper by itself.

I liked how you've deal with the issue of TC-climate feedbacks. As someone interested in them myself, I still feel that they are not completely proven. I therefore appreciated the way you had suitably couched your language on it. I remember that Prof. Kerty has written a parameterisation on this topic, but can't remember if it would capture the changes in genesis region.

Specific comments:

L16: unknow -> unknown

L68: please rephrase. I understand what you're conveying, but the [annual] mean temperature does reach 30oC in places in the Tropics in the present day.

L77: consider adding "in the early Eocene" to this sentence somewhere, as your previous sentence was talking about different climates in general.

L95: I don't think you should use the "increased (decreased)" style of writing here. It's not clear given the context what your two options are, esp as the previous sentence only has a single direction.

L122: Is it possible to add a further sentence talking about mechanisms behind the tropical meteorology you describe? For example, was the Walker Circulation substantially different in the Eocene? I recognise that doing additional analysis to test that is a unnecessary ask, but previous authors might have mentioned changes that could be a cause (e.g. Huber and Caballero).

L131: Is "concurrent" really justified. Figs 3b and 3f seem to show a different temporal evolution to me.

L145: Wouldn't "restriction" be better than "closure" for the Indonesian seaway?

L147: Please add "potential" to this sentence – you are not analysing actual TC formation.

L161: Again I'm not sure you should use the "increased (decreased)" style of writing here. The previous sentence has one option as "M.E.D rises", and this one talks about "larger (smaller) M.E.D arises"

L175-179: Please rephrase. I got lost in this long sentence.

L196: Please keep "most of the Cenozoic", but I do feel that "any of the Cenozoic" would also have been warranted.

Reviewer #2 (Remarks to the Author):

In this short paper, the authors report on a novel series of experiments that aims to deduce the tropical cyclone climate of different periods of the Cenozoic, along with the potential causes of these changes in climate. The paper is clearly written and the results appear new. I have some minor comments on aspects of the analysis and presentation.

Minor comments.

1. Line 30. It is not immediately obvious to the reader that the author is referring to the Early Eocene here when the phrase "during this time" is used.
2. There are a few places where the authors appear to make sweeping statements that are not entirely supported by their detailed results. This approach undoubtedly makes for a better story but in the interests of scientific accuracy, the authors may wish to consider the following:
 - a. Line 96: In the lower troposphere, the meridional temperature gradient shown in Fig. 2c does not appear to be systematically decreased between the two periods except poleward of 24° latitude. Thus this does not entirely constitute an explanation for the changes in low-level winds.
 - b. Line 98: In Fig. 2d, the moist entropy deficit is not increased everywhere: in fact, between 12°N and 20°N, it decreases. Also, there is no mention of the role of relative humidity in the changes in moist entropy deficit, despite relative humidity increasing at almost all latitudes in this panel.
 - c. Line 103: "deep tropics": the authors need to be systematic and specific about which latitudes are being referred to here.
 - d. Line 106: it is not necessarily the case that a larger vertical temperature gradient will lead to higher PI. For instance, simulations of future climate typically have higher PI but smaller vertical temperature gradient. The higher PI comes from an increase in the tropopause height and a subsequent decrease in the outflow temperature.
 - e. Line 107: for clarity, it is probably better to note that smaller vertical wind shear means a smaller vector difference in the winds between the upper and lower troposphere.
 - f. Line 109: the authors claim that Fig. 2h shows an increased vertical temperature contrast but almost all of the VTC numbers here are negative. Am I missing something?
 - g. Line 111: are these claimed mechanisms uniformly applicable over all of the panels found in Fig. S4? I haven't gone over every panel but based on the discussion of Fig. 2, I doubt it. The authors should point out if there are any specific parts of Fig. S4 where the results are unexpected.
 - h. Line 156: decreased PI does not occur everywhere in the South Indian Ocean in Fig. 4b. Vertical wind shear is not enhanced everywhere in the South Indian Ocean in Fig. 4d. Admittedly the area that does not agree with the authors statement is at rather low latitude and so is not likely to influence tropical cyclogenesis much, but this should at least be stated.
3. P. 8, description of the perturbation experiments. Can we please have a consistent description of these experiments that does not cause confusion? The experiments in question appear to be as follows: modern geography, modern geography with both Indonesia and Panama seaways open, and modern geography with just the Panama seaway open. So instead of sometimes talking about closing the seaways and sometimes about opening them, why not just talk either about opening them or closing them? The issue is that due to this confusion in the text, it took me a while to determine that Fig. 4 represented the differences for (modern) minus (modern with both Indonesia and Panama seaways open) e.g. in the nomenclature of Table S2, LP minus LP_openIP. So these plots show what happens if the seaways are closed, if I have got this right. I can see why this might be presented this way: geologically, the seaways gradually closed with time during the Cenozoic. But this does need to be carefully explained to avoid confusion.
4. Line 162, Fig. 4e. The difference between the colors of the dots is really hard to see. I suggest replacing the dots with a few selected contour lines.
5. Line 165: I'm not sure why a reduced meridional gradient of these environmental factors would necessarily cause an equatorward movement in GPI. Rather, the changes in GPI are simply due to geographical changes in the environmental variables.
6. Figure S9: since the experiments listed in Table S2 are only no seaway closure (LP_openIP),

both seaways closed (LP), and Indonesia seaway closed (LP_openP), I therefore think that Fig. S9 must be constructed by LP minus LP_openP, but it would be rather helpful to state that somewhere explicitly. Similarly, Fig. S7 must be LP_openP minus LP_openIP.

7. Line 203: the speculation about ocean mixing appears a bit out of place here, especially since no reference is given that backs up the authors' statement that this is a potential issue.

Text corrections.

1. Line 16: Should be "unknown".
2. Line 89: "stronger temperatures" should be "larger positive temperature anomalies".
3. Line 93: delete "positive" as this is confusing here.

Review of

Evolution of tropical cyclone genesis regions during the Cenozoic era

by Qing Yan and co-authors

(Reviewed by Kerry Emanuel)

The evolution of the climatology of tropical cyclones over geological time scales is not only of intrinsic interest but may have important feedbacks on the evolution of the climate itself. These feedbacks are currently not incorporated in coupled climate models as their resolution remains too coarse to simulate tropical cyclones with any accuracy. The present study uses a particular global coupled climate model to simulate the evolution of climate since the early Eocene period, and infers the genesis rates of tropical cyclones by applying a widely used “genesis potential index” (GPI) to the model output. Such indices have been shown to work well to simulate genesis rates in the current climate, including the effects of short-term climate fluctuations such as ENSO. The current application shows a remarkable evolution of genesis rates and locations over the Cenozoic era, with a strong migration of peak rates from the southern Indian Ocean to its current location in the western North Pacific. This should be of great interest to reader of *Nature Communications* and I recommend that it be accepted for publication, though I have a few suggestions for the authors.

One of the most interesting results is the apparently strong effect of changes in ocean circulation brought about by closing of seaways in Indonesia and Panama. The authors might consider some additional diagnostics related to the convergence of ocean heat flux.

On time scales longer than the those related to the thermal equilibration of the ocean mixed layer (roughly 2 years), the requirement of ocean mixed layer energy balance may be combined with the expression for potential intensity (Emanuel, 2007) to yield

$$V_p^2 = \frac{T_s - T_o}{T_o} \frac{F_{\downarrow} - F_{\uparrow} + F_{ocean}}{C_D \rho |\mathbf{V}|}, \quad (1)$$

where C_D is the surface drag coefficient, T_s is the sea surface temperature, T_o is the outflow temperature, ρ is an average air density near the surface, $|\mathbf{V}|$ is the average surface wind speed, F_{\uparrow} is the net infrared radiative flux out of the ocean, F_{\downarrow} is the net solar flux into the ocean, and F_{ocean} is the net energy flux into the ocean mixed layer by ocean-side processes, including entrainment.

One can see from this relation that the convergence of the ocean heat flux plays an important role in setting the potential intensity, especially in regions of low outflow temperature and light winds. It would be nice to use (1) as a diagnostic tool to understand why potential intensity changes, but this may require too much work for the present paper. On the other hand, perhaps it would not be too difficult to show maps of the convergence of ocean heat flux in the upper ocean, and this might really aid the interpretation of the results.

Specific comments:

1. Lines 89-90: “higher” rather than “stronger”, and “levels” rather than “level”.
2. Lines 95-98: This seems awkward. Why not infer stronger shear directly from increased temperature gradients rather than going through the unnecessary intermediary of lower and upper winds?
3. Lines 105-106: In potential intensity theory, the troposphere is always assumed to be nearly neutral to moist convection. Decreasing outflow temperature should be interpreted as either simply an increase in thermodynamic efficiency or an increase in the depth of the temperature perturbations. It has nothing to do with instability.
4. Lines 106-108: Same problem as mentioned above for lines 95-98.
5. Line 111-112: I find the sentence structure, involving the use of parentheses, to be confusing....please re-word.
6. General: My impression is that the closure of seaways accounts for most of the pattern of GPI change, but not for all of its amplitude. Is that correct? If so, it might be useful to state this.

Responses to Reviewer #1 (Dr. Chris Brierley)

I am impressed by this manuscript by Yan et al and feel that it is worthy of publication. They have undertaken a novel piece of research by combining the (sub)disciplines of palaeoclimatology and tropical meteorology to demonstrate an interesting finding about the earth system. I naturally have some questions and comments. But whether these are sufficient to require revisions, I will leave up to the editor to decide.

1. Methodologically, it builds upon established approaches and therefore is sound. NorESM-L is a climate model specifically built for past climate research. It was part of the Pliocene Model Intercomparison Project, so its fidelity at simulating warm climates can be assessed. It generally shows a similar behaviour to the other climate models. The use of a genesis potential index is also a well-established approach. Results can vary depending on the precise composition of the GPI, but the authors demonstrate that this is not a problem in this situation. I was somewhat surprised that there were not a few words comparing the NorESM-L late Pliocene simulated GPI change with the other PlioMIP models (from my 2015 paper). This is only because an Indian Ocean intensification only occurs in the MIROC5 model, and not the other 4 models we looked at (NorESM-L was not investigated in that paper). The late Pliocene is only a weak signal compared to the other setups testing here, so one would expect a weaker less robust signal. I do not want to see the point of model-dependency belaboured to the detriment of this interesting research. But since there is some pertinent evidence for it, I feel it should be acknowledged in the text.

Thank you very much for the helpful comments, and we agree with you that it is important to assess the robustness of our results. As you showed in Koh and Brierley (2015), the modeled GPI changes in the Late Pliocene can differ between models and basins rather significantly, but some features are at least qualitatively common across the ensemble: there is lower GPI during the Pliocene over the western North Pacific in most other PlioMIP models, and this is captured in NorESM-L too. It is true that only

MIROC shows substantially more favorable conditions at the Pliocene (relative to preindustrial era control) in the southern Indian Ocean in the PlioMIP models (Fig. 7 of Koh and Brierley), but our Figure 1 shows that the major shift away from the Indian Ocean to the Western North Pacific occurs earlier (in the transition from the Late Miocene to the Late Pliocene). There is qualitatively less difference between Late Pliocene and Modern in NorESM in the southern Indian Ocean, and this does not appear to us to be inconsistent with the ensemble mean difference or consistency (Fig. 8b, c of Koh and Brierley) with the other PlioMIP models.

More broadly, the reduced favorability for genesis at low latitudes that we find in the Early Eocene is also consistent with the downscaled TC behaviors from an Eocene-like climate simulation using CCSM (Korty et al., 2017). Although a more comprehensive comparison for other intervals of the Cenozoic is not feasible owing to the limited number of deep-time simulations that have been considered to date, in the revised manuscript (**lines 236-244**), we added some discussions related to what can be compared, and we thank you for pointing out the need to incorporate this in the text.

Koh, J. H., & Brierley, C. M. (2015). Tropical cyclone genesis potential across palaeoclimates. *Climate of the Past*, 11(10), 1433-1451.

Korty, R. L., Emanuel, K. A., Huber, M., & Zamora, R. A. (2017). Tropical cyclones downscaled from simulations with very high carbon dioxide levels. *Journal of Climate*, 30(2), 649-667.

2. My other substantive comment is that I wondered why you concentrate on the Indonesian Seaway as a being something like a tipping-point with an open vs closed state. Looking at the gradual reduction in GPI throughout the Cenozoic (Fig 3f), I got the impression this a more progressive process due the gradual incursion of Australasia into the Tropics. I suspect that there is a more to be found by looking at the tropical circulations, but this could easily form the basis of a paper by itself.

We acknowledge that the restriction of Indonesian Seaway is not a tipping-point for the formation of modern TC distribution over South Indian Ocean, and the evolution of GPI is a more progressive process. Following your suggestions, we examined the

variation of Hadley cell across the Cenozoic, which has been proven responsible for the poleward shift of TC activity in current warming world and past warm climates (Yan et al., 2016; Sharmila and Walsh, 2018). Our results illustrate that the Hadley cell extent gradually contracts equatorward in the Southern Hemisphere throughout the Cenozoic (Fig. R1; note that the variation of Hadley cell width is more complicated in the Northern Hemisphere), which contributes to the reduced GPI and the equatorward shift via modulating the large-scale genesis factors. However, a detailed analysis on tropical circulations is beyond the scope of current study and deserves more thorough investigation in future work.

In the revised manuscript (**lines 230-234**), we added: “*The Cenozoic evolution of genesis potential over the South Indian Ocean is a more progressive process than the abrupt transition that occurs in the western North Pacific, which may be linked with the gradual narrowing of Hadley cell in the Southern Hemisphere (Supplementary Figure 14) due to the northward incursion of Australia.*”

Figure R1. Zonal mean of the mass stream function (MSF) during the Cenozoic. The latitude where the MSF becomes zero at the poleward side of the subtropical maximum is defined as the boundary of the Hadley cell.

Sharmila, S., & Walsh, K. J. E. (2018). Recent poleward shift of tropical cyclone formation linked to Hadley cell expansion. *Nature Climate Change*, 8(8), 730-736.

Yan, Q., Wei, T., Korty, R. L., Kossin, J. P., Zhang, Z., & Wang, H. (2016). Enhanced intensity of global tropical cyclones during the mid-Pliocene warm period. *Proceedings of the National Academy of Sciences*, 113(46), 12963-12967.

3. I liked how you've deal with the issue of TC-climate feedbacks. As someone interested in them myself, I still feel that they are not completely proven. I therefore appreciated the way you had suitably couched your language on it. I remember that Prof. Korty has written a parameterisation on this topic, but can't remember if it would capture the changes in genesis region.

We agree that the case for a *significant* role for TC-ocean feedbacks of the type proposed more than a decade ago is incomplete, and indeed was somewhat undermined by later work arguing that much of the heat mixed into the thermocline is returned to the tropical atmosphere later in the annual cycle. (That is still a TC-ocean interaction, just one with somewhat different implications and consequences than those discussed in Korty et al. (2008), for example.) The parameterization developed in Korty et al. (2008) was based on some crude assumptions available at the time and scaled as a strong power of potential intensity (PI). Although the justification they used for scaling mixing with a high power of PI relied on speculation that power dissipation might increase in warm climates owing to (1) stronger storms and (2) perhaps more of them, work in the last decade has provided much more information about TCs and climate than was known then: the first point is well supported, but the evidence for (2) is complicated and mixed. (Most models make fewer total storms in warmer climates, but many yield a larger number of very strong storms.) Nevertheless, a concurrent empirical estimate of power dissipation—which is the physical basis for the mixing—developed in Emanuel (2007) from observations also supports a strong, high-exponent dependence of power dissipation on PI. Here we briefly consider what each of these formulations imply about possible TC mixing, with the caveat that we feel this issue would profit from revisiting the work done a decade ago in light of new TC-climate lessons.

a) Following the methods of Korty et al. (2008), TC-induced vertical mixing broadly scales with the potential intensity and the storm contribution to ocean mixing can be estimated as:

$$\kappa_s(x, y) = \kappa \left[\frac{PI(x, y)}{PI_*} \right]^6, \quad (1)$$

where κ_s is the diffusion coefficient caused by TCs, κ is a constant and is set to $0.1 \text{ cm}^2/\text{s}$, PI is the potential intensity, and PI_* is set to 70 m/s here. The estimated the TC-induced vertical mixing using this formula can be easily applied in climate models.

b) Power dissipation is related to the amount of wind energy available to mix the upper ocean, and hence determines the magnitude of TC-induced vertical mixing. Following the method of Emanuel (2007), power dissipation is estimated as:

$$PDI \sim |\eta|^{5/2} PI^7 (1 + 0.3VS)^{-4}, \quad (2)$$

where PI is the potential intensity, VS is the vertical wind shear between 200 and 850 hPa, and $|\eta|$ is the absolute vorticity at 850 hPa.

c) Although the estimated effect of TCs on vertical mixing varies with different methods (e.g., Eq. 1 vs. Eq. 2) and has large uncertainties, they are consistent in that they both suggest TC-induced vertical mixing would vary greatly across the climates in the Cenozoic (Fig. R2). In the revised manuscript, these results are added to the supplementary information (i.e., **Supplementary Figures 16, 17**).

Figure R2. Estimated effect of TC activity on (a) ocean diffusion coefficient based on Korty et al. (2008) and (b) power dissipation based on Emanuel (2007).

Korty, R. L., Emanuel, K. A., & Scott, J. R. (2008). Tropical cyclone–induced upper-ocean mixing and climate: Application to equable climates. *Journal of Climate*, 21(4), 638-654.

Emanuel, K. (2007). Environmental factors affecting tropical cyclone power dissipation. *Journal of Climate*, 20(22), 5497-5509.

Specific comments:

4. L16: unknow -> unknown

Corrected.

5. L68: please rephrase. I understand what you're conveying, but the [annual] mean temperature does reach 30oC in places in the Tropics in the present day.

Rephrased to “simulated annual sea surface temperatures (SSTs) averaged over the tropics (30°S–30°N)”

6. L77: consider adding “in the early Eocene” to this sentence somewhere, as your previous sentence was talking about different climates in general.

Done.

7. L95: I don't think you should use the “increased (decreased)” style of writing here. It's not clear given the context what your two options are, esp as the previous sentence only has a single direction.

Rephrased.

8. L122: Is it possible to add a further sentence talking about mechanisms behind the tropical meteorology you describe? For example, was the Walker Circulation substantially different in the Eocene? I recognise that doing additional analysis to test that is a unnecessary ask, but previous authors might have mentioned changes that could be a cause (e.g. Huber and Caballero).

One of the most obvious changes in GPI between the early Eocene and present-day is that the location of genesis regions was poleward of its present-day location during the early Eocene (i.e., reduced GPI at low latitudes in the Eocene with an increase at higher (subtropical) latitudes). This meridional shift may be associated with change in the Hadley cell as suggested by previous studies (Yan et al., 2016; Sharmila and Walsh, 2018). Our results show that the general structure of Hadley circulation holds unchanged in the early Eocene compared with the present-day (Fig. R3), but there is a poleward expansion of the Hadley circulation in the early Eocene, consistent with the poleward migration of high GPI. In the revised manuscript (lines 138-140), we added some discussions on this point.

Figure R3. The mass stream function (MSF; $\text{kg/s} \times 10^9$) in the (a) Early Eocene, (b) the present, and (c) its anomaly between the Early Eocene and present.

Sharmila, S., & Walsh, K. J. E. (2018). Recent poleward shift of tropical cyclone formation linked to Hadley cell expansion. *Nature Climate Change*, 8(8), 730-736.

Yan, Q., Wei, T., Korty, R. L., Kossin, J. P., Zhang, Z., & Wang, H. (2016). Enhanced intensity of global tropical cyclones during the mid-Pliocene warm period. *Proceedings of the National Academy of Sciences*, 113(46), 12963-12967.

9. L131: Is “concurrent” really justified. Figs 3b and 3f seem to show a different temporal evolution to me.

We deleted the word “concurrent”.

10. L145: Wouldn't “restriction” be better than “closure” for the Indonesian seaway?

Modified as suggested.

11. L147: Please add “potential” to this sentence – you are not analysing actual TC formation.

Done.

12. L161: Again I'm not sure you should use the “increased (decreased)” style of writing here. The previous sentence has one option as “M.E.D rises”, and this one talks about “larger (smaller) M.E.D arises”

Rephrased.

L175-179: Please rephrase. I got lost in this long sentence.

Done.

L196: Please keep “most of the Cenozoic”, but I do feel that “any of the Cenozoic” would also have been warranted.

Done.

Responses to Reviewer #2

In this short paper, the authors report on a novel series of experiments that aims to deduce the tropical cyclone climate of different periods of the Cenozoic, along with the potential causes of these changes in climate. The paper is clearly written and the results appear new. I have some minor comments on aspects of the analysis and presentation.

We sincerely thank the reviewer for the helpful comments, and reply to each of them below.

Minor comments.

1. Line 30. It is not immediately obvious to the reader that the author is referring to the Early Eocene here when the phrase “during this time” is used.

Changed to “during the Early Eocene”.

2. There are a few places where the authors appear to make sweeping statements that are not entirely supported by their detailed results. This approach undoubtedly makes for a better story but in the interests of scientific accuracy, the authors may wish to consider the following:

Thank you very much for these very detailed and helpful comments.

2a. Line 96: In the lower troposphere, the meridional temperature gradient shown in Fig. 2c does not appear to be systematically decreased between the two periods except poleward of 24° latitude. Thus this does not entirely constitute an explanation for the changes in low-level winds.

Thank you very much for pointing this out. In the revised manuscript, we used the changes in meridional temperature gradient over the troposphere to explain the variations of vertical wind shear, instead of going through the intermediary of lower and upper level winds. This relies on thermal wind arguments that an increase in temperature gradient appears in concert with an increase in vertical wind shear, and

vice versa. In the updated Fig. 2c, we show that changes in wind shear closely match the variation of temperature gradient through the depth of the troposphere. For example, the decreased wind shear at the north of $\sim 22^\circ\text{N}$ is caused by the smaller meridional temperature gradient over that region.

Figure 2c. Differences in vertical wind shear between 200 and 850 hPa (VS; m s^{-1}) and absolute meridional temperature gradient (MTG; $\times 10^{-6} \text{ }^\circ\text{C m}^{-1}$) in the troposphere between the Early Eocene and pre-industrial.

2b. Line 98: In Fig. 2d, the moist entropy deficit is not increased everywhere: in fact, between 12N and 20N, it decreases. Also, there is no mention of the role of relative humidity in the changes in moist entropy deficit, despite relative humidity increasing at almost all latitudes in this panel.

In the revised manuscript (**lines 114-119**), we added: *“The moist entropy deficit anomaly broadly exhibits a tripole pattern, with a decrease at the zonal band of $\sim 12\text{--}20^\circ\text{N}$ and an increase along its northern and southern sides (Fig. 2d). The increased moist entropy deficit is largely caused by a decreased temperature contrast between the mid-troposphere and the surface, which weakens the strength of the surface heat fluxes, whereas the smaller deficit between $12\text{--}20^\circ\text{N}$ results from increased relative humidity (Fig. 2d).”*

2c. Line 103: “deep tropics”: the authors need to be systematic and specific about which latitudes are being referred to here.

We defined “deep tropics” as the zonal band of ~5–15°N, and explicitly state this in the revised manuscript (now line 121).

2d. Line 106: it is not necessarily the case that a larger vertical temperature gradient will lead to higher PI. For instance, simulations of future climate typically have higher PI but smaller vertical temperature gradient. The higher PI comes from an increase in the tropopause height and a subsequent decrease in the outflow temperature.

Corrected. On a related note, we have also added material on the causes behind PI changes following the partitioning first discussed in Emanuel (2007): changes in thermodynamic efficiency (i.e., outflow and surface temperature differences), net surface fluxes, ocean heat convergence, and near surface wind speeds. In the revised manuscript (**lines 104-110 and updated Fig. 2**), we added this analysis to aid the interpretation of changes in potential intensity.

Emanuel, K. (2007). Environmental factors affecting tropical cyclone power dissipation. *Journal of Climate*, 20(22), 5497-5509.

2e. Line 107: for clarity, it is probably better to note that smaller vertical wind shear means a smaller vector difference in the winds between the upper and lower troposphere.

Rephased as suggested.

2f. Line 109: the authors claim that Fig. 2h shows an increased vertical temperature contrast but almost all of the VTC numbers here are negative. Am I missing something?

We corrected the text as follows: “*The decreased moist entropy deficit over the majority of South Indian Ocean results from the larger relative humidity, though the vertical temperature contrast is also decreased (Fig. 2h).*”

2g. Line 111: are these claimed mechanisms uniformly applicable over all of the panels found in Fig. S4? I haven't gone over every panel but based on the discussion of Fig. 2, I doubt it. The authors should point out if there are any specific parts of Fig. S4 where the results are unexpected.

We acknowledge that the changes in GPI and the associated dominant factors vary for different storm basins. In the revised manuscript (**lines 134-137**), we added additional discussions on this point following your suggestions.

2h. Line 156: decreased PI does not occur everywhere in the South Indian Ocean in Fig. 4b. Vertical wind shear is not enhanced everywhere in the South Indian Ocean in Fig. 4d. Admittedly the area that does not agree with the authors statement is at rather low latitude and so is not likely to influence tropical cyclogenesis much, but this should at least be stated.

In the revised manuscript (**lines 182-203**), changes at very low latitudes of the southern Indian Ocean are explicitly discussed. For example, we added: *“Each of these factors contributes to a more favorable environment over the western North Pacific, but a less favorable one in the South Indian Ocean (except at very low latitudes where GPI is increased) when the tropical seaways are closed (Fig. 4g), and there is an overall equatorward migration of genesis regions.”*

3. P. 8, description of the perturbation experiments. Can we please have a consistent description of these experiments that does not cause confusion? The experiments in question appear to be as follows: modern geography, modern geography with both Indonesia and Panama seaways open, and modern geography with just the Panama seaway open. So instead of sometimes talking about closing the seaways and sometimes about opening them, why not just talk either about opening them or closing them? The issue is that due to this confusion in the text, it took me a while to determine that Fig. 4 represented the differences for (modern) minus (modern with both Indonesia and Panama seaways open) e.g. in the nomenclature of Table S2, LP minus LP_openIP. So these plots show what happens if the seaways are closed, if I

have got this right. I can see why this might be presented this way: geologically, the seaways gradually closed with time during the Cenozoic. But this does need to be carefully explained to avoid confusion.

Thank you very much for the helpful comments. In the revised manuscript (**lines 178-182**), we renamed these experiments and added additional information in the text to avoid confusion. For example, we wrote: *“Taking the Late Pliocene experiment with closed Indonesian and Panama seaways as the baseline (referred to as LP_closeIP), in the first experiment we keep the Indonesian seaway closed but open the Panama seaway (referred to as LP_closeI); in a second experiment (referred to as LP_noclosure), we open both tropical seaways.”*

4. Line 162, Fig. 4e. The difference between the colors of the dots is really hard to see. I suggest replacing the dots with a few selected contour lines.

Done.

5. Line 165: I’m not sure why a reduced meridional gradient of these environmental factors would necessarily cause an equatorward movement in GPI. Rather, the changes in GPI are simply due to geographical changes in the environmental variables.

Agreed. We mean that the anomalies of genesis factors generally show a dipole pattern over the South Indian Ocean, with more favorable conditions for storm formation at the low latitudes of tropical ocean and less favorable conditions at the relatively high latitudes. In the revised manuscript, we rephrased as: *“Each of these factors contributes to a more favorable environment over the western North Pacific, but a less favorable one in the South Indian Ocean (except at very low latitudes where GPI is increased) when the tropical seaways are closed (Fig. 4g), and there is an overall equatorward migration of genesis regions.”*

6. Figure S9: since the experiments listed in Table S2 are only no seaway closure (LP_openIP), both seaways closed (LP), and Indonesia seaway closed (LP_openP), I

therefore think that Fig. S9 must be constructed by LP minus LP_{openP}, but it would be rather helpful to state that somewhere explicitly. Similarly, Fig. S7 must be LP_{openP} minus LP_{openIP}.

Modified as suggested (please also see our response to comment#3).

7. Line 203: the speculation about ocean mixing appears a bit out of place here, especially since no reference is given that backs up the authors' statement that this is a potential issue.

In the revised manuscript, we cited several references that support the potential role of TC-induced ocean mixing in global/tropical climate, and provided a rough estimation on the effect of TCs on upper ocean vertical mixing (i.e., Supplementary Figures 16, 17).

Text corrections.

1. Line 16: Should be "unknown".

Corrected.

2. Line 89: "stronger temperatures" should be "larger positive temperature anomalies".

Corrected.

3. Line 93: delete "positive" as this is confusing here.

Done.

Responses to reviewer #3 (Prof. Kerry Emanuel)

A. The evolution of the climatology of tropical cyclones over geological time scales is not only of intrinsic interest but may have important feedbacks on the evolution of the climate itself. These feedbacks are currently not incorporated in coupled climate models as their resolution remains too coarse to simulate tropical cyclones with any accuracy. The present study uses a particular global coupled climate model to simulate the evolution of climate since the early Eocene period, and infers the genesis rates of tropical cyclones by applying a widely used “genesis potential index” (GPI) to the model output. Such indices have been shown to work well to simulate genesis rates in the current climate, including the effects of short-term climate fluctuations such as ENSO. The current application shows a remarkable evolution of genesis rates and locations over the Cenozoic era, with a strong migration of peak rates from the southern Indian Ocean to its current location in the western North Pacific. This should be of great interest to readers of *Nature Communications* and I recommend that it be accepted for publication, though I have a few suggestions for the authors.

One of the most interesting results is the apparently strong effect of changes in ocean circulation brought about by closing of seaways in Indonesia and Panama. The authors might consider some additional diagnostics related to the convergence of ocean heat flux.

On time scales longer than the those related to the thermal equilibration of the ocean mixed layer (roughly 2 years), the requirement of ocean mixed layer energy balance may be combined with the expression for potential intensity (Emanuel, 2007) to yield

$$V_p^2 = \frac{T_s - T_o}{T_o} \frac{F_{\downarrow} - F_{\uparrow} + F_{ocean}}{C_D \rho |V|}$$

where C_D is the surface drag coefficient, T_s is the sea surface temperature, T_o is the outflow temperature, ρ is an average air density near the surface, $|V|$ is the average surface wind speed, F_{\uparrow} is the net infrared radiative flux out of the ocean, F_{\downarrow}

is the net solar flux into the ocean, and F_{ocean} is the net energy flux into the ocean mixed layer by ocean-side processes, including entrainment.

One can see from this relation that the convergence of the ocean heat flux plays an important role in setting the potential intensity, especially in regions of low outflow temperature and light winds. It would be nice to use (1) as a diagnostic tool to understand why potential intensity changes, but this may require too much work for the present paper. On the other hand, perhaps it would not be too difficult to show maps of the convergence of ocean heat flux in the upper ocean, and this might really aid the interpretation of the results.

Thank you very much for the very insightful comments.

Based on your suggestions, we use the formulae that follow to examine factors controlling changes in potential intensity in our simulations. The algorithm used to compute PI follows the development of Bister and Emanuel (2002) in comparing the CAPE of a parcel lifted from saturation at the surface with the CAPE of a boundary layer parcel. An alternate way to express PI uses the thermodynamic disequilibrium at the surface expressed in terms of enthalpy:

$$PI = \sqrt{\frac{C_k}{C_d} \frac{SST - T_o}{T_o} (k_0^* - k)}, \quad (1)$$

where T_o is the mean outflow temperature, C_k is the exchange coefficient for entropy, C_d is the drag coefficient, k_0^* is the enthalpy of air saturated at the sea surface temperature and pressure, and k is the enthalpy of an ambient boundary layer parcel. But when SSTs (and mixed-layer heat content) are in steady-state, the thermodynamic disequilibrium (i.e., $k_0^* - k$) can be related to surface and ocean fluxes through surface energy balance:

$$k_0^* - k = \frac{F_- - F_+ + F_{ocean}}{C_k r |V|}, \quad (2)$$

Substituting (2) into (1) yields an alternate expression for PI, first presented as eqn. (7) of Emanuel (2007):

$$PI = \sqrt{\frac{SST - T_o}{T_o} \frac{F_{\downarrow} - F_{\uparrow} + F_{ocean}}{C_d \rho |V|}}, \quad (3)$$

Thus, four quantities can directly affect PI: (i) the thermodynamic efficiency term involving differences in SST and outflow temperature T_o (i.e., at the level of neutral buoyancy or the equilibrium level); (ii) net surface fluxes; (iii) convergence of ocean heat flux, and (iv) the near-surface wind speed.

We have analyzed the component terms in Eq. 3 to attempt to assess why PI differs between the open seaway and closed cases. We applied this to annual mean values of PI and other quantities (not the storm season means), as the assumption of steady-state conditions was challenging to apply on subannual timescales. In any event, PI exhibits little annual cycle equatorward of 20° latitude, at least in the western Pacific and Indian Oceans, which are our primary regions of interest.

For the near-surface wind speed, we assume log-layer theory and use geopotential height data of the lowest model level to extrapolate to an estimate of 10-m velocities. Also, note all quantities are computed from long-term means, without consideration of the interannual variability in them.

Regarding the experiments targeting the role of tropical seaways, we find that the difference in potential intensity when tropical seaways are closed is strongly affected by changes in ocean heat convergence and near-surface wind speeds (Fig. R1). Note that here we have calculated the ocean heat convergence in Eq. 3 as a residual, as all other variables are available from the atmospheric model and PI algorithm directly. We verified this is qualitatively similar to what the ocean heat flux constructed from ocean model data, but that chart is choppier and subject to assumptions about mixed-layer depth (which we don't have archived), so we present ocean heat convergence (F_{ocean}) here as a residual.

There are two main regions where changes in F_{ocean} appear to dominate the change in PI are (1) the equatorial Pacific and (2) southern Indian Ocean between 10°S and 20°S. Closing the Indonesian (and Panamanian) seaways leads to higher ocean heat convergence in the Pacific, leading to higher SST and PI. In the Indian Ocean, ocean heat convergence is larger between 10°S and 20°S when the seaway is

open, and SST and PI are correspondingly higher here in that case. Note, however, that changes in the near-surface wind speed are also important, and appear to control the response from the equatorial Indian Ocean eastward to the Pacific side of Indonesia. This is made somewhat more explicit by squaring Eq. 3 and taking the natural logarithm of the equation, to attribute changes in PI to individual terms (Fig. R2). This follows the technique in your earlier papers, but with the additional use of the relationship

$$\ln(F_{net} + F_{ocn}) = \ln \left(F_{net} \left(1 + \frac{F_{ocn}}{F_{net}} \right) \right) = \ln F_{net} + \ln \left(1 + \frac{F_{ocn}}{F_{net}} \right), \quad (4)$$

to (mostly) separate the influence of ocean fluxes from the net short and longwave surface flux ($F_{net} = F_{\downarrow} - F_{\uparrow}$); panels R2c and R2d show the difference in the two terms on the right hand side of (4), respectively.

Regarding the Early Eocene, we show that, for example, the decreased potential intensity at low latitudes (5–12°N) of the western North Pacific is mainly caused by the reduced air-sea disequilibrium relative to present (Fig. R3) resulting from weakened surface radiative flux and enhanced surface wind speed (Fig. R4), whereas it is largely attributed to higher outflow temperatures at higher latitudes which leads to smaller thermodynamic efficiency given larger warm temperature anomalies in the upper troposphere than the surface.

In the revised manuscript (**lines 104-140&182-203&343-374**), we added more analysis to better interpret the changes in genesis factors according to your helpful comments.

Figure R1. Differences in annual mean environmental factors caused by the tropical seaway closures (LP_closeIP minus LP_noclosure). (a) potential intensity (m s^{-1}), (b) SST (K), (c) outflow temperature (K), (d) surface wind speed (m s^{-1}), (e) net surface radiative fluxes (net downward shortwave radiation minus net upward longwave radiation; W m^{-2}); (f) ocean heat convergence (positive values indicate anomalous heat convergence; W m^{-2}) calculated as a residual.

Figure R2. Differences in the logarithm of annual mean environmental factors caused by the tropical seaway closures (LP_closeIP minus LP_noclosure). (a) shows the change in $2 \cdot \ln (PI)$, and (b-d) that follow are the four pieces that add to it (see text for details).

Figure R3. Differences in zonal mean GPI and environmental variables during storm season between the Early Eocene and pre-industrial averaged over the western North Pacific (110–160°E). (a) GPI (number of events per month). (b) Potential intensity (PI; m s^{-1}), the enthalpy difference between sea surface and boundary layer ($k^* - k$; J kg^{-1}), and the outflow temperature (OT; °C). (c) Vertical wind shear between 200 and 850 hPa (VS; m s^{-1}) and absolute meridional temperature gradient (MTG; $\times 10^{-6} \text{ } ^\circ\text{C m}^{-1}$) in the troposphere. (d) Moist entropy deficit (X), vertical temperature contrast (VTC; °C) between surface and mid-troposphere (600 hPa), and relative humidity (RH; %) at 600hPa.

Figure R4. Differences in environmental factors during storm season averaged over the western North Pacific (110–160°E) between the Early Eocene and pre-industrial. (a) enthalpy difference between sea surface and boundary layer ($k^* - k$; J kg^{-1}), (b) net radiative flux at the sea surface (W m^{-2}), (c) the convergence of upper ocean heat flux (W m^{-2} ; estimated as a residual), and (d) surface wind speed (m s^{-1}).

Specific comments:

1. Lines 89-90: “higher” rather than “stronger”, and “levels” rather than “level”.

Corrected.

2. Lines 95-98: This seems awkward. Why not infer stronger shear directly from increased temperature gradients rather than going through the unnecessary intermediary of lower and upper winds?

Following your suggestions, in the revised manuscript, we used the changes in meridional temperature gradient over the troposphere to explain the variations of vertical wind shear. And we found that changes in wind shear closely match the variation of temperature gradient (e.g., Fig. R3c).

3. Lines 105-106: In potential intensity theory, the troposphere is always assumed to be nearly neutral to moist convection. Decreasing outflow temperature should be interpreted as either simply an increase in thermodynamic efficiency or an increase in the depth of the temperature perturbations. It has nothing to do with instability.

This point is corrected based on your suggestions (please also see our responses to comment#A for details).

4. Lines 106-108: Same problem as mentioned above for lines 95-98.

Modified as suggested.

5. Line 111-112: I find the sentence structure, involving the use of parentheses, to be confusing...please re-word.

Rephrased.

6. General: My impression is that the closure of seaways accounts for most of the pattern of GPI change, but not for all of its amplitude. Is that correct? If so, it might be useful to state this.

We acknowledge that the closure of tropical seaways accounts for most of the pattern of GPI change between the Late Miocene and Late Pliocene, but the amplitude of GPI change is underestimated. This may be associated with the setup of the depth and width of tropical seaways. In our experiments, the Indonesian seaway is broadened by converting the northern part of New Guinea (11 grid cells) to ocean with a depth of ~50 m; and the Panama seaway is opened by removing one land grid cell and setting the depth of new ocean grid cell to 25 m. As the modeled climate responses depends on the depth and width of tropical seaways (Zhang et al.2012; Sepulchre et al., 2013), we may expect a larger response if we increase the depth and width of tropical seaways applied here. In the revised manuscript (**lines 223-228**), the underestimation of GPI amplitude is clearly stated, and we added some discussions on this point.

Zhang, X., Prange, M., Steph, S., Butzin, M., Krebs, U., Lunt, D. J., ... & Schulz, M. (2012). Changes in equatorial Pacific thermocline depth in response to Panamanian seaway closure: Insights from a multi-model study. *Earth and Planetary Science Letters*, 317, 76-84.

Sepulchre, P., Arsouze, T., Donnadieu, Y., Dutay, J. C., Jaramillo, C., Le Bras, J., ... & Waite, A. J. (2014). Consequences of shoaling of the Central American Seaway determined from modeling Nd isotopes. *Paleoceanography*, 29(3), 176-189.

REVIEWERS' COMMENTS:

Reviewer #2 (Remarks to the Author):

The authors have satisfactorily addressed my comments.

Reviewer #3 (Remarks to the Author):

The authors have done a superb job addressing my first review (and the other two reviews as well) and I appreciate their careful responses to my earlier points. I particularly appreciate their highlighting the changes made to the manuscript from the original version.

I find the current version acceptable for publication and think it will make a valuable contribution to the understanding of the relationship between climate change and tropical cyclones.

Kerry Emanuel